# The multi-photon induced Fano effect

K. L. Litvinenko [1✉], Nguyen H. Le [1], B. Redlich [2], C. R. Pidgeon[3], N. V. Abrosimov[4], Y. Andreev[5,6], Zhiming Huang[7] & B. N. Murdin [1]

The ordinary Fano effect occurs in many-electron atoms and requires an autoionizing state. With such a state, photo-ionization may proceed via pathways that interfere, and the characteristic asymmetric resonance structures appear in the continuum. Here we demonstrate that Fano structure may also be induced without need of auto-ionization, by dressing the continuum with an ordinary bound state in any atom by a coupling laser. Using multi-photon processes gives complete, ultra-fast control over the interference. We show that a line-shape index $q$ near unity (maximum asymmetry) may be produced in hydrogenic silicon donors with a relatively weak beam. Since the Fano lineshape has both constructive and destructive interference, the laser control opens the possibility of state-selective detection with enhancement on one side of resonance and invisibility on the other. We discuss a variety of atomic and molecular spectroscopies, and in the case of silicon donors we provide a calculation for a qubit readout application.

[1] Department of Physics, Advanced Technology Institute, University of Surrey, Guildford GU2 7XH, UK. [2] FELIX Laboratory, Institute for Molecules and Materials, Radboud University, Nijmegen, The Netherlands. [3] Institute of Photonics and Quantum Science, SUPA, Heriot-Watt University, Edinburgh, UK. [4] Leibniz-Institut für Kristallzüchtung (IKZ), Berlin, Germany. [5] Institute of Monitoring of Climatic and Ecological Systems of SB RAS, 10/3, Academicheskii Avenue, Tomsk 634055, Russia. [6] National Research Tomsk State University, 1, Novosobornaya Strasse, Tomsk 634050, Russia. [7] State Key Laboratory of Infrared Physics and Laboratory of Space Active Opto-Electronics Technology, Shanghai Institute of Technical Physics, CAS, 500 Yutian Road, Shanghai 200083, China. ✉email: k.litvinenko@surrey.ac.uk

Fano resonance structures occur when a discrete state and a continuum are coupled, creating two pathways from the ground state to the continuum that interfere[1]. They are found in a very wide variety of physical situations including atomic physics[2,3], nano-mechanics[4], condensed matter and semiconductor physics[5–10], and photonics[11–14]. Their characteristic asymmetric lineshape is extremely useful because of the structure it gives to an otherwise flat continuum and the sharp change that it affords from transmission to reflection, or from scattering to invisibility. For example, continuum structure can be used to make laser mirrors with very desirable switching properties[14]. In the classic atomic Fano process, a continuum is coupled to a discrete state by an 'auto-ionizing' interaction, i.e., an energy-conserving configuration interaction such as an electron–electron interaction in helium[1] in which one electron relaxes and the other is ejected. In almost all examples of Fano processes, this internal interaction is a constant, but sometimes the shape index (the Fano $q$-parameter) may be chosen by sample design[4,10], or controlled by temperature[9] or external magnetic field[7]. The structure may also be controlled by the intensity if it is very strong and non-perturbative[8]. Very exciting and flexible possibilities for control occur if the coupling interaction with the continuum is instead optically induced[15]. In this situation, not only can the Fano resonance be controlled by the coupling beam intensity, the discrete state need not be auto-ionizing, allowing continuum structure to be obtained in principle for any atom at any point in the continuum. Until now such Induced Continuum Structure (ICS) has only been achieved with extremely high field-amplitude coupling beams[16–19], using a process that gives control only over the width, not the shape index. Using multi-photon transitions, both the Fano shape index and width are controllable, and here we show that by making use of a THz-coupling beam the ICS can be obtained with a relatively weak beam. We used hydrogenic donor impurities in silicon, which have very large multi-photon cross-sections[20,21] producing a shape index close to unity for an easily accessible field amplitude of only a few 10s of kV/cm.

Many-electron atoms possess discrete states in which each electron remains localized, but the total energy of all the electrons is enough to ionize one of them. An example is the $|a\rangle = |2s, 2p\rangle$ state in helium[1] and its semiconductor analogue the double donor[5], which can be produced with a single photon excitation from the $|g\rangle = |1s, 1s\rangle$ ground state (it is dipole allowed) and can spontaneously decay to $|k_\omega\rangle = |1s, c\rangle$, which comprises one electron in the 1s state and one in the continuum (i.e., one relaxing and one being ejected). This decay is called auto-ionization, because it costs no extra energy ($|a\rangle$ is above the one-electron continuum edge) and the interaction that produces it is a static configuration interaction such as the internal electron–electron interaction. As shown in Fig. 1a (in the cartoon labelled Fano), photo-ionization from $|g\rangle$ can therefore proceed either via $|a\rangle$ or by direct promotion of one electron to $|c\rangle$, and these pathways interfere producing Fano resonance structures in the spectrum[1]. In effect, the interaction, $V$, between the discrete state and the continuum (the electron–electron configuration interaction in the helium example just given), dresses $|c\rangle$ with $|a\rangle$, and the phase of this mixture changes sign either side of $|a\rangle$ producing constructive and destructive interference when probed with an optical dipole transition with perturbation interaction, $D$. Fano structures appear for Multi-Photon Ionization (MPI) where the optical probe $D$ is replaced by $D^{(N)}$[22], which may be Resonance-Enhanced[23] (labelled RE-MPI in Fig. 1a), and may also involve multiple intermediate states and colours[24,25]. These examples—and indeed almost all examples of electronic Fano resonances—involve auto-ionizing states with positive energy. The excited states of hydrogenic atoms are bound, i.e., they are not auto-

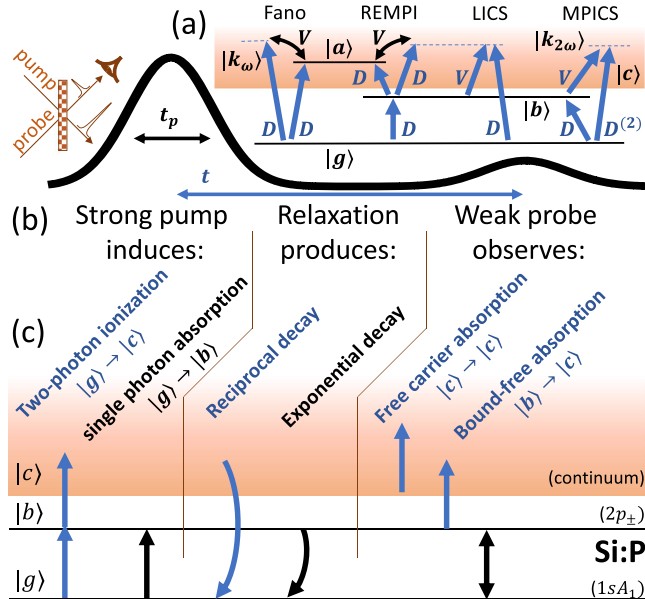

**Fig. 1 Fano and related processes. a** Here, $|g\rangle$ is the ground state, $|c\rangle$ (the orange band) is the one-electron continuum, and $|a\rangle$ is an auto-ionizing state, i.e., it is discrete and localized, but it has enough total energy that one electron can be kicked out of the atom into $|c\rangle$. $|k_\omega\rangle$ (dashed blue line) is the state within the continuum that is resonant with excitation from $|g\rangle$ by dipole allowed transitions with perturbation Hamiltonian $D$ at frequency $\omega$, indicated here such that $|k_\omega\rangle$ is above the level of $|a\rangle$, i.e., positive detuning, $\Delta$. $V$ is an energy-conserving perturbation (such as a configuration interaction) that couples $|a\rangle$ with $|c\rangle$, and creates a second pathway from $|g\rangle \rightarrow |k_\omega\rangle$ via $|a\rangle$ and the pathways interfere. As the detuning $\Delta$ is tuned across the resonance with $|a\rangle$, the interference between the pathways changes sign. Resonance-Enhanced Multi-Photon Ionization (RE-MPI) is similar to the Fano process, involving an intermediate state. In Laser-Induced Continuum Structure (LICS), $V$ is a dipole transition from a bound state $|b\rangle$ and now $|g\rangle \rightarrow |k_\omega\rangle$ interferes with a third-order process $|g\rangle \rightarrow |k\rangle \rightarrow |b\rangle \rightarrow |k_\omega\rangle$, where $|k\rangle$ is a virtual intermediate state and no autoionizing state is involved. In Multi-Photon Induced Continuum Structure (MPICS), interference occurs between two second-order processes $|g\rangle \rightarrow |b\rangle \rightarrow |k_{2\omega}\rangle$ and $|g\rangle \rightarrow |k\rangle \rightarrow |k_{2\omega}\rangle$, where the latter is a two-photon transition via a virtual intermediate state with Hamiltonian $D^{(2)}$. In degenerate MPICS (used in this work), $V = D$. **b** The pump-probe experiment geometry in time and space. The pulse duration $t_p$ and delay time $t$ are indicated. **c** The primary processes in a standard pump-probe experiment (black) and the additional ionization-related processes (blue) of interest here.

ionizing, and therefore Fano lineshapes would not normally be expected. However, new Fano structures have been predicted[15] when the interaction between the discrete state and the continuum is induced by a coupling laser, and the discrete state may then be any ordinary bound state, $|b\rangle$, which makes Fano resonances much more widely applicable. The simplest of these ICS processes are shown in Fig. 1a: Laser-ICS (LICS), a kind of $\Lambda$-scheme with two laser beams[16–19] where each transition to the continuum indicated takes one photon, and Multi-Photon-ICS (MPICS) in which each step involves two photons and may be degenerate.

The Fano line-shape of width $\Gamma$ and shape index $q$ is

$$f(\Delta; \Gamma, q) = \frac{(\Delta + q\Gamma)^2}{\Delta^2 + \Gamma^2} \qquad (1)$$

where $\Delta = \hbar(\omega - \omega_0)$ is the detuning of the probe from resonance. $f$ has a peak of height $1 + q^2$ at $\Delta = \Gamma/q$ and also a zero at

$\Delta = -q\Gamma$. The zero disappears if $q \to \infty$, leaving only a symmetric resonance, and the peak disappears if $q \to 0$, leaving only a symmetric anti-resonance. If the interfering pathways contribute equally then the shape index $q$ is near unity and this situation is useful because it produces both an enhancement and a window. Selected species or structures might therefore be ionized, whereas others are invisible to the ionizing beam, enabling, e.g., isotope separation (as we discuss below).

As mentioned earlier, control over the continuum structure is desirable. In LICS although $q$ is fixed, the width is intensity controllable even in the perturbative limit: $\Gamma \propto F^2$, where $F$ is the field amplitude of the coupling laser. In MPICS, both $\Gamma \propto F^2$ and $q \propto 1/F^2$ are controllable[15]. The attraction of ICS in general is now clear: it may purely involve stable bound states available in any atom and the Fano effect can be switched on and off at will (on ultrafast timescales if the coupling beam is ultrafast). In the case of MPICS in particular, only one beam is needed and the shape index is controllable. The intensities required for ICS are generally very high and an optical control over the line-shape index has never been observed experimentally even though MPICS was first proposed more than 40 years ago. Just as silicon double donors are, in many respects, semiconductor analogues of free helium atoms and exhibit the classic Fano auto-ionization structures[5], so silicon shallow donors are analogues of free hydrogen atoms. They exhibit a reduced Hartree energy $E_H$ and an increased Bohr radius $a_B$[26,27], and this scaling tends to enhance nonlinear optical effects and reduce the intensities needed[20,21]; hence, their interest for ICS here.

Signatures of photo-ionization have frequently been observed in silicon donors[28–30] (and acceptors[31,32]) as an unwanted arte-fact when performing pump-probe experiments. Here we examine this process more closely in phosphorus-doped silicon (same sample as in ref. [27]). A schematic of the experiment is shown in Fig. 1b (for more details, see 'Methods').

## Results

**Temporal and spectral dependence of photo-bleaching and -ionization.** We shall mainly be concerned with the transition from $|g\rangle = |1s\rangle$ to $|b\rangle = |2p_\pm\rangle$ at $\hbar\omega = 39.2$ meV. The pump-induced change in probe transmission as a function of the time delay is shown in Fig. 2 for a variety of intensities and wavelengths on or close to this resonance. At the lowest intensity on resonance, the transient is positive (induced bleaching) and the recovery is exponential with time constant $T_1 = 93 \pm 1$ ps. The dynamics are very different for off-resonance excitation and a clear intensity-dependent induced absorption (negative induced transmission) is visible, especially on the red side of resonance (Fig. 2).

The small-signal absorption spectrum of the sample is shown in Fig. 3a, showing the hydrogenic Lyman series. To illustrate the spectral dependence of the laser experiment more clearly, we fixed the time delay (at $t = 15$ ps) and swept the photon energy, as shown in Fig. 3b. Strongly asymmetric features are evident, with negative signals (induced absorption) on the red side, notably for $1s \to 2p_0$ and $1s \to 2p_\pm$. These negative signals are also visible in Fig. 2b, which was taken near the $1s \to 2p_\pm$ transition with the laser red-shifted to the point that its spectrum does not overlap with the small-signal absorption (blue curve on Fig. 3a), close to the minimum in Fig. 3b. The induced absorption effect disappears on the timescale of the electronic relaxation back to the donor ground state, which rules out any thermal or purely optical interference effects.

**Extraction of probe absorption coefficients and decay constants.** We now wish to extract from Fig. 2c the frequency

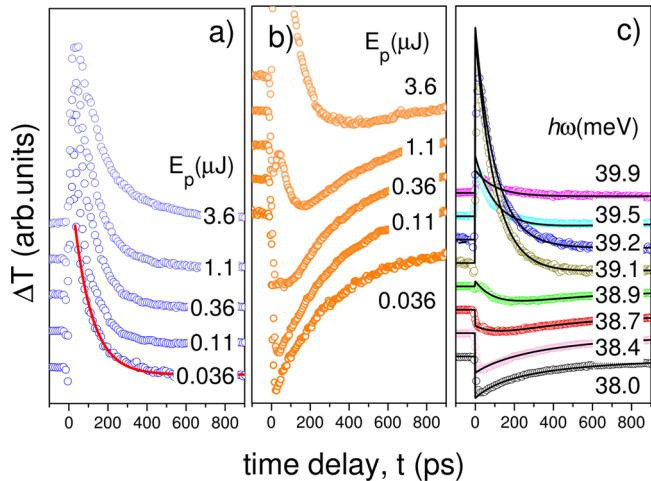

**Fig. 2 Pump-probe transients on or near the $1s$–$2p_\pm$ transition for different laser intensities.** The transient transmission change of the probe due to the pump pulse, $\Delta T$ is shown vs. $t$. They have been normalized to their global maximum and offset vertically. **a** Pump pulse energy ($E_p$) dependence on resonance (at laser photon energy $\hbar\omega = 39.2$ meV) for $^{nat}$Si:P. The solid line is an exponential decay fit. **b** As in **a** but off-resonance at $\hbar\omega = 38.6$ meV. **c** Dependence of the transients on photon energy ($\hbar\omega$) at a fixed pulse energy of $E_p = 0.11 \mu$J. The solid lines are fits with Eq. (3) as described in the text.

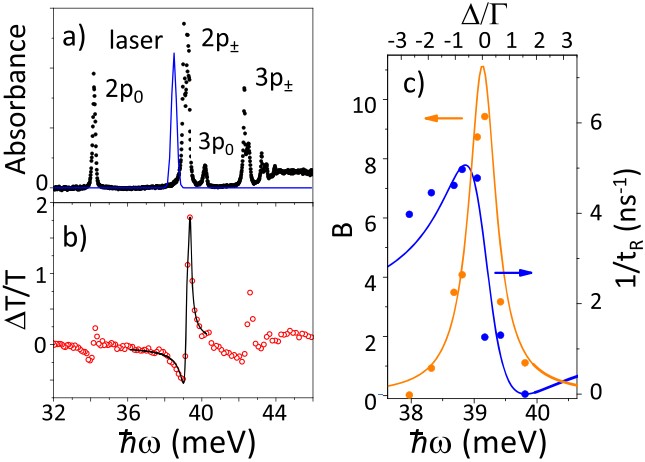

**Fig. 3 Spectral dependence of induced transmission and continuum structure with the Fano profile. a** The small-signal absorbance spectrum. The main transitions are labelled according to the excited states (all originating from the 1s(A₁) ground state). The laser spectrum corresponding to the off-resonance photon energy used in Fig. 2c is shown as a blue line. **b** Wavelength dependence of the pump-probe signal, $\Delta T/T$, with fixed pump power (same as in Fig. 2c) and fixed time delay of $t = 15$ ps between pump and probe as a function of photon energy, $\hbar\omega$. The solid curve is the result of a fit to the data using Eq. (1) as described in the Methods. **c** The recombination rate (blue symbols), $1/t_R$, and bleaching (orange symbols), $B$, found from fitting the data in Fig. 2c. The solid curves are the results of a global fit using Eq. (1) for the ionization (blue line) and a Lorentzian for the bleaching (orange line). The top axis shows the detuning $\Delta$ in units of the width from the fit, $\Gamma$.

dependence of the ionization density produced by the two-photon pumping, by separating the effects of the excitation/ bleaching/relaxation from the photo-ionization/induced absorp-tion/recombination[30] (see 'Methods'). As illustrated in Fig. 1c, the pump generates excitation from $|g\rangle$ to $|b\rangle$ and two-photon

ionization from $|g\rangle$ to $|c\rangle$. The population in the excited state, $n_b$, follows a rate equation of the form $dn_b/dt = -n_b/T_1$, which has an exponential decay solution $n_b(t) = n_b(0)\exp(-t/T_1)$, where the time constant is $T_1$. The recombination of free electrons with ionized donors follows a rate equation of form $dn_c/dt = -Rn_cn_i = -Rn_c^2$ where $n_c$ is the density of free electrons and $n_i$ is the density of ions, and $n_i = n_c$ by charge conservation. $R$ is the recombination rate coefficient[33]. This differential equation has so-called reciprocal decay solutions of the form $n_c(t) = n_c(0)(1 + t/t_R)^{-1}$ where

$$t_R = 1/Rn_c(0) \tag{2}$$

is the recombination time. The excited population is responsible for induced bleaching and the free electrons produce free-carrier absorption (FCA). The resulting differential transmission has the form (see 'Methods')

$$\Delta T = Be^{-t/T_1} - C(1 + t/t_R)^{-1} + A \tag{3}$$

The coefficients $B$ and $C$ describe the degree of excitation produced by the pump to $|b\rangle$ and $|c\rangle$, respectively. The constant term $A$ represents a long timescale decay produced by free electrons becoming trapped in other defects with very slow return. We performed fits of Eq. (3) to the data of Fig. 2c (shown by the solid lines), in which we treated the relaxation time $T_1$ as a global constant. The best fit value of $T_1$ was $102 \pm 4$ ps, which is within $2\sigma$ of the value given earlier for the local fit of the lowest intensity data on Fig. 2a. The three dimensionless transmission change coefficients ($A$, $B$, $C$) and the recombination timescale ($t_R$) depend on the degree of excitation by the pump, which varies with intensity and photon energy. The latter four parameters (i.e., not including $T_1$) were allowed to vary for each trace, and the resulting values for $B$ and $1/t_R(\omega)$ are shown in Fig. 3c. It is noteworthy that the maximum of $1/t_R$ is on the red side of the line on Fig. 3c, and because this scales with the induced ionization and therefore also with the induced FCA, it corresponds to the minimum in the induced transmission on Fig. 3b.

**Fano spectrum of photo-ionization due the pump.** From Eq. (2), the inverse recombination time is simply proportional to the ionized electron density left behind by the pump and it is clear from the $1/t_R$ data (Fig. 3c) that the ionization is strongly asymmetric with a Fano-like spectrum. We therefore wish to extract the Fano shape index and width. It can be shown that $B = (2\sigma_{gb} - \sigma_{bc})n_b(0)L$ and $C = (\sigma_{cc} - \sigma_{gb})n_c(0)L$ (see 'Methods') where $L$ is the sample thickness (Fig. 1c right), $\sigma_{gb}$ is the absorption cross-section for the resonant excitation from $|g\rangle \rightarrow |b\rangle$, $\sigma_{bc}$ is the cross-section for photo-ionization from the excited state, and $\sigma_{cc}$ is the FCA cross-section. The bound-free and free-free cross-sections, $\sigma_{bc,cc}$, are slowly varying with frequency (the standard Drude model for FCA produces $\sigma_{cc} \propto \omega^{-2}$) and we take them to be constant over the range of each feature in Fig. 3b, c. We further assume that bound-free absorption is weak compared with resonant absorption, which simplifies $B$. It is clear from the $B$ data (Fig. 3c) that the excitation has a simple, symmetric, resonant spectrum, and so we assume that $\sigma_{gb}$ and the excited density $n_b(0)$ have a Lorentzian spectrum of the same half-width $\gamma$. As mentioned above, we assume that $n_c(0)$ is given by Eq. (1). With these assumptions,

$$\begin{aligned}
t_R^{-1}(\omega) &= t_r^{-1}f(\omega - \omega_0; \Gamma, q) \\
B(\omega) &= \beta l^2(\omega - \omega_0; \gamma), \\
C(\omega) &= \chi[1 - \chi_\sigma l(\omega - \omega_0; \gamma)]f(\omega - \omega_0; \Gamma, q)
\end{aligned} \tag{4}$$

where $l(\Delta; \gamma)$ is the normalized Lorentzian, $\omega_0$ is the line centre, $\beta$, $\chi$, $\chi_\sigma$, $t_r$ are global constants simply related to $L$ and the values at

the centre of the transition of $\sigma_{cc}$, $\sigma_{gb}$, $n_b(0)$, and $n_c(0)$. To extract these parameters, we performed a simultaneous fit to the spectrum in Fig. 3b, the $1/t_R$, $B$ data in Fig. 3c, and the $C$ data. All the parameters in Eq. (4) were taken to be global constants and we minimized the average value of the r.m.s. residual for each of the four functions, each r.m.s. residual being weighted by the respective r.m.s. signal. From this fit, the line centre was $\hbar\omega_0 = 39.13 \pm 0.01$ meV (c.f. the accepted value for $1s \rightarrow 2p_\pm$ of 39.175 meV[26]), $\hbar\gamma = 0.26 \pm 0.02$ meV, $\hbar\Gamma = 0.43 \pm 0.12$ meV, and $q = -1.6 \pm 0.1$. The experimental blue shift of the zero is therefore $\hbar\Delta_0 = -\hbar q\Gamma = 0.69$ meV.

**Multi-photon ICS.** We now consider the Fano theory[1] and the results just obtained for $q$. The classic Fano process is shown in Fig. 1a. The three states $|g\rangle, |a\rangle, |c\rangle$ are stationary states without the interaction $V$, which mixes a component of $|a\rangle$ into $|c\rangle$ producing a dressed state $|\Psi\rangle$. The transition rate from $|g\rangle \rightarrow |\Psi\rangle$ produced by the dipole interaction $D = e\mathbf{F}.\mathbf{r}.\varepsilon$ for probe light with frequency $\omega$, field amplitude $F$, and polarization $\varepsilon$ may be found from the Golden Rule. The relevant matrix element is $\langle g|d|\Psi\rangle \propto \frac{\Delta}{E_H}\langle g|d|k_\omega\rangle + \langle g|d|a\rangle\langle a|v|k_\omega\rangle + \langle g|dG_\omega^{(k_\omega)}v|a\rangle\langle a|v|k_\omega\rangle$ where $d = \mathbf{r}.\varepsilon/a_B$, $v = V/E_H$, and $|k_\omega\rangle \in |c\rangle$ is the unmixed continuum state at energy $E_g + \hbar\omega$, i.e., resonant with the excitation.

The three terms in the matrix element $\langle g|d|\Psi\rangle$ indicate the interfering pathways, $|g\rangle \xrightarrow{D} |k_w\rangle, |g\rangle \xrightarrow{D} |a\rangle \xrightarrow{V} |k_w\rangle$, and the last one is the total contribution via off-resonant states, $|g\rangle \xrightarrow{D} |k \neq k_w\rangle \xrightarrow{V} |a\rangle \xrightarrow{V} |k_w\rangle$ produced by

$$G_\omega^{(k_\omega)} = E_H \sum_{k \neq k_\omega} \frac{|k\rangle\langle k|}{E_k - E_g - \hbar\omega}. \tag{5}$$

The $\Sigma$ symbol in $G_\omega^{(k_\omega)}$ includes both the summation over the discrete states and the integration over the continuum states. As $k_\omega$ is in the continuum, the exclusion $k \neq k_\omega$ should be understood as taking the principal value of the integration part. The rate for $|g\rangle \rightarrow |\Psi\rangle$ divided by the background rate to the unmixed continuum, $|g\rangle \rightarrow |k_\omega\rangle$, may be written with the form of Eq. (1) with $q = \langle g|d|a\rangle/[\pi\langle g|d|k_\omega\rangle\langle k_\omega|v|a\rangle]$ and $\Gamma = \pi E_H|\langle a|v|k_\omega\rangle|^2$. For a flat continuum, both $q$ and $\Gamma$ are independent of $\omega$.

In our experiment, $\omega$ is near to resonance with an allowed transition to a bound state and ionization requires two photons, as in MPICS[15] (Fig. 1a). We used a single laser, so the induced Fano-coupling interaction is the one-photon perturbation produced by the same beam ($V = D$). The MPICS rate is found from the Golden Rule generalized for second-order processes, i.e., it now includes the Kramers–Heisenberg–Goeppert matrix element[34], $\langle g|(d - dG_\omega^{(b)}d)|\Psi\rangle \propto \langle g|d|b\rangle\langle b|d|k_{2\omega}\rangle - \frac{\Delta}{E_H}\langle g|dG_\omega^{(b)}d|k_{2\omega}\rangle$ neglecting a term that is the fourth order in $d$. The pathways are therefore $|g\rangle \rightarrow |b\rangle \rightarrow |k_{2\omega}\rangle$ and $|g\rangle \rightarrow |k \neq b\rangle \rightarrow |k_{2\omega}\rangle$. Now the ionization rate is

$$R(\Delta) = \frac{\pi E_H}{2\hbar}|\langle g|dG_\omega^{(b)}d|k_{2\omega}\rangle|^2 \frac{F^4}{F_a^4}f(\Delta; \Gamma, q) \tag{6}$$

where

$$q = -\frac{\langle g|d|b\rangle}{\pi\langle g|dG_\omega^{(b)}d|k_{2\omega}\rangle\langle k_{2\omega}|d|b\rangle}\frac{F_a^2}{F^2}, \tag{7}$$

$$\Gamma = \pi E_H|\langle b|d|k_{2\omega}\rangle|^2 F^2/F_a^2, \tag{8}$$

and $F_a = E_H/ea_B$ is the Coulomb field at the Bohr radius. See 'Methods' for theoretical details. Eqs. (7) and (8) show that the shape is controllable with field.

**Implicit summation of the non-resonant matrix element.** The matrix element for the pathway via off-resonant intermediate states produced by $D^{(2)}$, i.e., $\langle g|dG_\omega^{(b)}d|k_{2\omega}\rangle$ appearing in Eqs. (6) and (7), has the form $\langle\psi|d|k_{2\omega}\rangle$, where $|\psi\rangle = G_\omega^{(b)}d|g\rangle$. The sum inside the operator (Eq. (5)) appearing here is over an infinite number of intermediate states. In a similar way to the problem of two-photon absorption between bound states, implicit summation[20,21,35] can reduce this difficulty substantially. In this technique $|\psi\rangle$ becomes the solution of partial differential equation (PDE) that does not require construction of $G_\omega^{(b)}$. Here we used a new variant (see 'Methods'), which takes account of the excluded state. In this case, $|\psi\rangle$ is the solution of

$$\left(H_0 - E_g - \hbar\omega\right)|\psi\rangle = E_H(\mathbb{1} - |b\rangle\langle b|)d|g\rangle \qquad (9)$$

where $H_0$ is the Hamiltonian without the interactions ($D$, $D^{(2)}$), the eigenstates of which are $|g\rangle$, $|b\rangle$ etc.

The calculation for $|b\rangle = |2p\rangle$ in hydrogen produces $\langle g|d|b\rangle = 0.74$, $\langle b|d|k_{2\omega}\rangle = 0.38$, $\langle g|dG_\omega^{(b)}d|k_{2\omega}\rangle = 8.5$, which are all of order unity, and therefore it is apparent from Eq. (7) that to achieve $q$ near unity $F$ must be of order $F_a$. There have been no previous experimental observations of a Fano lineshape in MPICS, because $F_a = 5.1$ GV/cm; thus, unless the laser is extremely intense, $q$ is very large and the resonance reduces to very narrow Lorentzian.

As a first approximation for Si:P, the hydrogen result may be scaled to the hydrogenic donor using $E_H = e^4m/\hbar^2\kappa^2$ and $a_B = \hbar^2\kappa/e^2m$ (where the symbols have their usual meaning) and inserting the appropriate dielectric constant and the effective mass. We use no other parameters (adjustable or otherwise). Taking $a_B = 3.2$ nm and $E_H = 40$ meV[27] produces $F_a = 130$ kV/cm, so defining $q = -F_1^2/F^2$ and $\Delta_0 = -q\Gamma$, where $F_1$ is the field required to achieve unity $|q|$ and $\Delta_0$ is the blue shift of the Fano window (which is independent of field), we find unity $q$ expected at $F_1 = 36$ kV/cm and the zero is expected at $\hbar\Delta_0 = 1.4$ meV.

## Discussion

From above, the experimental blue shift of the zero is $\hbar\Delta_0 = 0.69$ meV, within about a factor of 2 from the prediction just given. The pump pulse energy and duration used in the experiments of Figs. 2c and 3b, c give an estimated experimental half-maximum electric field amplitude of $F = 22$ kV/cm. Therefore, we find experimentally $F_1 = \sqrt{|q|}F = 28$ kV/cm, which compares very well with theory. Put another way, using the experimental $F$ in the theoretical formula for $q$ predicts $q = -2.7$, which is also within a factor of two of experiment. These values have some significant uncertainty: the experimentally fitted value of $\Gamma$ has an error of about 30%, and there is also potentially large systematic uncertainty in the experimental field amplitude due to the problem of pulse temporal profile estimation. Furthermore, the hydrogen scaling theory does not take account of the valley multiplicity, mass anisotropy or central cell correction[21,26]. The effect of these theoretical corrections is illustrated by looking at the $2p_0$ transition, as the scaled hydrogen model would predict that all components of the $2p$ would have the same $q$ and $\Gamma$. Although the $2p_0$ feature in Fig. 3b is too small to fit reliably, it is clear that its experimental $q$ and $\Gamma$ (and $\Delta_0$ are different from those of the $2p_\pm$, although only slightly, which is consistent with small differences in the matrix elements caused by the mass anisotropy, etc. Considering these uncertainties and approximations, the agreement everywhere within a factor of 2 is very good.

State-selective ionization and detection is a crucial step in quantum state readout for spin-based qubit devices with dot readout fidelities of over 99%[36–40]. Till now, field ionization for resonant tunnelling readout has been at low or very low frequency[36–38,40], and high-frequency THz photon-assisted ionization of impurities has been either strong-field but non-resonant[41,42], or using weak (and therefore inefficient) donor-bound exciton transitions[39,43]. Resonant THz-assisted charge transfer between interacting donors when only one of the pair is excited has also been reported[28]. The Fano resonances demonstrated here have ultrafast switch-on times in the few-ps range (as shown in Fig. 2), and even recovery times that are comparable with the inverse of the typical microwave spin qubit Larmor frequencies (e.g., X-band at 10 GHz), and much faster than consequent Bloch rotation times[36–40]. For concreteness we give a specific example for Si:Bi, a popular qubit choice[44–46], which displays large zero-field spin splitting (originating from hyperfine interaction) visible in the small-signal absorption shown on Fig. 4. There is therefore a Fano ionization profile for each spin, and blue-shifting the laser to the window of one uniquely ionizes the other. Figure 4 shows the example for the $3p$ transition, for which $\langle g|d|b\rangle = 0.30$, $\langle b|d|k_{2\omega}\rangle = 0.14$, $\langle g|dG_\omega^{(b)}d|k_{2\omega}\rangle = 4.2$. In this example, the ionization rate of the lower spin level when the laser is at the window of the upper level (red minimum on the diagram) is 10 GHz when the field amplitude is 20 kV/cm, i.e., allowing essentially 100% readout fidelity with a nanosecond pulse. We note that in this type of application the fact that $q\Gamma$ is independent of field amplitude makes the window robust against intensity fluctuations and controlling $q$ with the field amplitude controls the ionization rate of the other spin.

The implications are wider than qubit readout schemes, since high cross-section bound states are available in many solid-state systems and atomic/molecular vapours. In particular, very interesting possibilities may be afforded by non-degenerate, two colour experiments. Utilizing a THz pulse (similar to that used here) to couple the continuum to Rydberg states in ordinary atoms and a visible 'probe' beam for excitation from the ground to Rydberg state also produces very strong, useful Fano structures. In this case, $F$ appearing in $q$ and $\Gamma$ (Eqs. (7) and (8)) is replaced by the coupling beam field $F_c$, and in the rate (Eq. (6)) $F$ is replaced by $\sqrt{F_cF_p}$, the geometric mean of coupling and probe fields. In each, $k_{2\omega}$ is replaced by $k_{\omega_c+\omega_p}$. The advantage relative to degenerate excitation with a visible beam is that $k_{\omega_c+\omega_p}$ is much closer to the continuum edge and much less quickly varying, so the matrix elements appearing in the denominator of $q$ are very much larger, easily compensating for any reduction in the numerator. Thus, $F_1$ is much smaller and unity $q$ is much more easily attainable. For example, we consider the case of Rb and the $40p$ Rydberg state at ~4.14 eV above the ground state and a coupling beam provided by a THz quantum cascade laser (QCL) diode (e.g., 1.2 kW in a 550 ns pulse at 3.38 THz[47], photon energy 14 meV). Using the Whittaker quantum defect wavefunction for the ground state of Rb[48,49] and the hydrogenic wavefunction for the Rydberg state, we obtain $\langle g|d|b\rangle = 0.0099$, $\langle b|d|k_{\omega_c+\omega_p}\rangle = 340$, and $\langle g|dG_{\omega_p}^{(b)}d|k_{\omega_c+\omega_p}\rangle = 2.8\times 10^5$. One obtains $F_1 \approx 30$ kV/cm, i.e., very much less than $F_a$, in stark contrast with the degenerate hydrogen $2p$ example from above. Unity $q$ could thus be obtained if the QCL mentioned were focussed to a spot of four wavelengths in diameter. This rapidly spectrally varying continuum structure, induced by THz lasers, opens the possibility of controllable ionization and scattering for a variety of state-selective or isotope-specific ionization applications. The fact that the ICS is available in a many ordinary vapours will allow a variety of new atomic and molecular physics spectroscopies.

## Methods

**Experiment.** The silicon sample was cut in the [100] direction from a crystal grown by the float zone technique. The P-donor concentration was near $n_p = 2 \times 10^{15}$ cm$^{-3}$ (same sample as in ref. [27]). The small-signal absorbance spectrum of Fig. 3a was taken at 4.2 K using Fourier transform infrared spectrocopy with 0.01

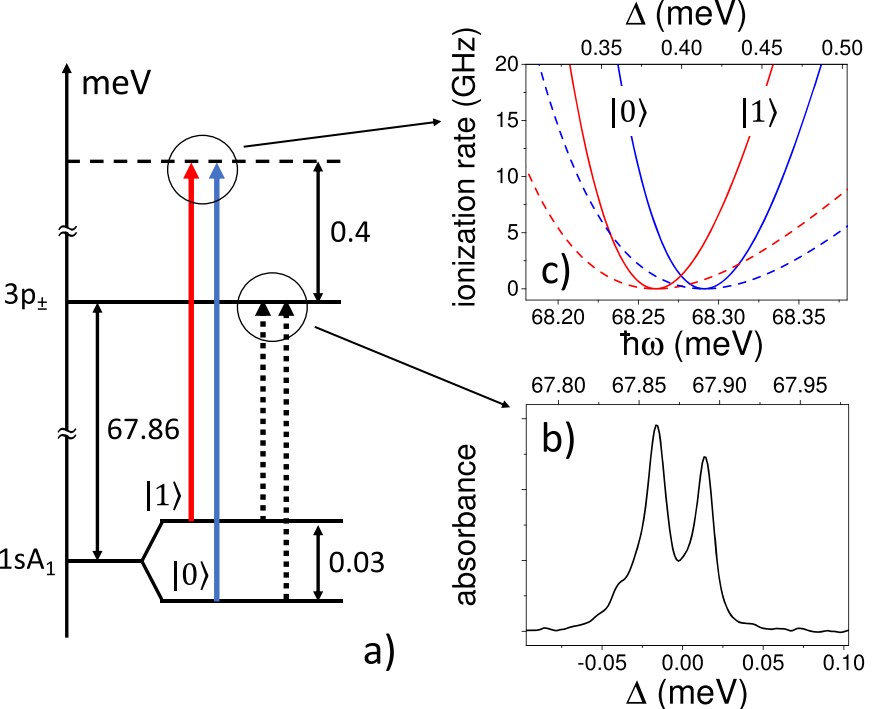

**Fig. 4 An application of multi-photon induced continuum structure. a** The level scheme of Si:Bi with a hyperfine split ground state and the $3p_\pm$ excited state. The MPICS ionization window is located at a detuning $\Delta = 0.4$ meV above resonance. **b** Small-signal absorption spectrum against photon energy $\hbar\omega$ of Si:Bi illustrating the well-resolved hyperfine splitting (data from ref. [50]). **c** The MPICS windows for a field amplitude in atomic units of $F/F_a = 0.15$ (solid line) and 0.1 (dash line) illustrating the available state-selective ionization contrast.

meV resolution. The laser used was the Free Electron Laser FELIX, which emits tunable picosecond pulses of high intensity. Its spectrum is approximately Gaussian, and the central frequency and r.m.s. width were recorded during the experiment with a spectrum analyser. An example synthetic spectrum reconstructed from this is shown in Fig. 3a. The laser repetition frequency was 25 MHz.

The optical setup was a standard pump-probe experiment. For a detailed description of this arrangement, see refs. [28–30]. The sample was mounted in vacuum on a cryostat cold-finger at 5 K. THz pulses from a Free Electron Laser were focussed through room-temperature, thin-film windows. The r.m.s. radius of the beam intensity was 0.5 mm. The intensity of the probe beam was kept 15 times less than the pump. The small-signal absorption spectrum of the sample is shown in Fig. 3a. The half-width of each absorption line is ~0.15 meV, which was therefore somewhat narrower than the laser spectrum half-width of 0.19 meV used in the pump-probe experiment. The laser is near bandwidth-limited, so the pulses have a half-width duration of $t_p = 2.5$ ps.

**Data analysis.** The rate equations for densities undergoing the relaxation processes shown in Fig. 1c (middle) are:

$$\frac{dn_b}{dt} = -\frac{n_b}{T_1} + \rho R n_c^2 \tag{10}$$

$$\frac{dn_c}{dt} = -R n_c^2 \tag{11}$$

where the total rate for recombination of ions with free electrons is $R$. An extra term is included in the first of these, to take account of the possibility of recombination directly into the excited state, the proportion of which is $\rho$. The solutions are

$$n_c = \frac{n_{c0}}{1 + t/t_R} \tag{12}$$

$$n_b = (n_{b0} + \rho n_{c0})e^{-\frac{t}{T_1}} - \frac{\rho n_{c0}}{1 + t/t_R} + \rho n_{c0}\zeta\left(\frac{t}{T_1}, \frac{t_R}{T_1}\right) \tag{13}$$

where $\zeta(z, z_R) = -z_R e^{-z - z_R} \int_{-z - z_R}^{-z_R} x^{-1} e^{-x} dx$ and $t_R = 1/Rn_{c0}$ is the recombination time. $n_{b0}$ and $n_{c0}$ are the values of $n_b$ and $n_c$ at $t = 0$.

The effect of the pump is governed by non-classical processes, but the probe is weak, and (away from $t = 0$ when it is simultaneous with the pump) its absorption may be described classically. The probe transmission for a sample of thickness $L$ is

$$T = \exp(-\sigma_{gb}(n_g - n_b)L - \sigma_{bc}n_b L - \sigma_{cc}n_c L). \tag{14}$$

The three terms in the exponential correspond to the three processes shown in Fig. 1c (right): $n_{g,b,c}$ are the densities of ground state donors, excited bound state donors, and ions, and $\sigma_{gb}$ is the cross-section for excitation from $|g\rangle \rightarrow |b\rangle$, etc. The total density is $n_D = n_g + n_b + n_c$, i.e., we neglect the density in all other states, and in equilibrium at $t = \pm\infty$ $n_b = n_c = 0$. Therefore, for small thickness, the change induced by the pump is

$$\Delta T = T - T_\infty \approx \sigma_{gb}[2n_b + n_c]L - \sigma_{bc}n_b L - \sigma_{cc}n_c L$$
$$= (1 + \rho)Be^{-t/T_1} - \frac{C + \rho B}{1 + t/t_R} + \rho B\zeta\left(\frac{t}{T_1}, \frac{t_R}{T_1}\right) + A \tag{15}$$

where $B = (2\sigma_{gb} - \sigma_{bc})n_{b0}L$, $C = (\sigma_{cc} - \sigma_{gb})n_{c0}L$. We have added a phenomenological constant term, $A$, to represent ionized electrons that become trapped in other defects with a slow release.

We now have a functional form for the transients of Fig. 2. It is described by six independent parameters: three dimensionless transmission coefficients ($A$, $B$, $C$), two timescales ($t_R$, $T_1$), and the fraction of recombination that occurs into the excited state ($\rho$). Of these, $T_1$ and $\rho$ are expected to be constants of the system, while the other four depend on $n_{c0,b0}$, i.e., on the degree of excitation by the pump, which varies with intensity and photon energy. We therefore fitted each of the transients in Fig. 2c with $T_1$, $\rho$ as global variables, while $A$, $B$, $C$, $t_R$ were different for each trace. We found that $\rho = 0$, in which case $\Delta T$ reduces to Eq. (3).

**Implicit summation theory.** The calculation of $q$ in Eq. (7) requires $\langle g|dG_\omega^{(b)}d|k_{2\omega}\rangle$. Let $d|g\rangle = |\psi'\rangle$ and $\langle g|dG_\omega^{(b)}d|k_{2\omega}\rangle = \langle \psi|d|k_{2\omega}\rangle$. Therefore, $|\psi\rangle = G_\omega^{(b)}|\psi'\rangle$. By inspection,

$$E_H^{-1}\left(H_0 - E_g - \hbar\omega\right)|\psi\rangle = \sum_{k \neq b}|k\rangle\langle k|\psi'\rangle$$
$$= (1 - |b\rangle\langle b|)|\psi'\rangle = |\psi'\rangle - |b\rangle\langle b|\psi'\rangle \tag{16}$$

because of the completeness of the $|k\rangle$'s. The second term on the right hand side of the above PDE has arisen because of the excluded state. The PDE can be solved to extract $|\psi\rangle$ without trying to construct $G_\omega^{(b)}$ explicitly. We used the Finite Element Method for our calculation. At resonance $E_g + \hbar\omega = E_b$, which is an eigenenergy of $H_0$, causing numerical instability in the inversion $(H_0 - E_b)^{-1}$. In order to overcome this we solve the PDE with $E_b$ replaced by $E_b + I\eta$ and take the limit $\eta \rightarrow 0$.

**Parameters of the MPICS lineshape.** In the main text, we present our result with a semi-classical description of the field. This allows a simple and intuitive connection between the original Fano resonance and the multi-photon induced Fano

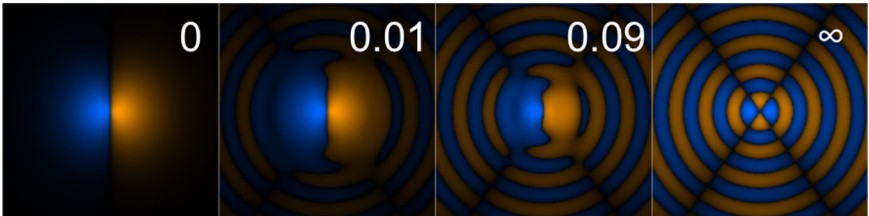

**Fig. 5 The MPICS wavefunction.** $\Psi(\mathbf{r}) \propto b(\mathbf{r}) + \theta k_{2\omega}(\mathbf{r})$ is shown for $|b\rangle \equiv |2p_\pm\rangle$ in hydrogen, in the $xz$ plane, with $z$ chosen to be the axis of the field polarization, at different values of $|\theta|$. For $\theta = 0$, $\Psi(\mathbf{r}) \propto b(\mathbf{r})$, while for $\theta = \infty$, $\Psi(\mathbf{r}) \propto k_{2\omega}(\mathbf{r})$. The colour indicates the sign, and the brightness indicates the amplitude, normalized to the maximum in each case.

resonance. Here we detail the formal derivation of the formulae, which is best done when the field is treated as quantum[15].

The Hamiltonian for the system of an atom interacting with a laser mode with frequency $\omega$ is

$$H = H_0 + \hbar\omega\, a_\omega^\dagger a_\omega + H_{\mathrm{I}}, \tag{17}$$

where

$$H_{\mathrm{I}} = i\left(\frac{\hbar\omega}{2\epsilon_0\epsilon_r L^3}\right)^{1/2} e\, \mathbf{r}.\boldsymbol{\epsilon}\left(a_\omega - a_\omega^\dagger\right) \tag{18}$$

is the interaction Hamiltonian, $\epsilon_r$ the dielectric constant of silicon, $L^3$ the quantization volume and $a_\omega$ the annihilation operator.

Let there be initially $n$ photons in the field and let $\omega$ be close to the resonance between $|g\rangle$ and $|b\rangle$. The initial atom-field state $|g, n\rangle$ can be excited to the continuum state $|\kappa_{2\omega}, n-2\rangle$ by either a sequential single photon excitation through $|b, n-1\rangle$ or a direct two-photon excitation process through the non-resonant intermediate states $|j \neq b, n-1\rangle$. The normalization for $|\kappa_{2\omega}\rangle$ is as usual for continuum states, $\langle\kappa|\kappa'\rangle = \delta(E_\kappa - E_{\kappa'})$, so that $|\kappa_{2\omega}\rangle$ is related to $|k_{2\omega}\rangle$ in the main text by $|k_{2\omega}\rangle = \sqrt{E_{\mathrm{H}}}|\kappa_{2\omega}\rangle$. As the discrete state $|b, n-1\rangle$ has the same energy as the continuum state $|\kappa_{2\omega}, n-2\rangle$ and is coupled to it by $H_{\mathrm{I}}$, it can be seen as a pseudo-autoionizing state[15]. This coupling produces the dressed Fano state[1,15]:

$$|\chi\rangle = \frac{1}{\sqrt{\Delta^2 + \Gamma^2}} \times \left(P_{b,n-1}H_{\mathrm{I}} + \Delta + \mathcal{G}_{2\omega}^{(\kappa_{2\omega})}H_{\mathrm{I}}P_{b,n-1}H_{\mathrm{I}}\right)|\kappa_{2\omega}, n-2\rangle, \tag{19}$$

where

$$\begin{aligned} \Gamma &= \pi|\langle b, n-1|H_{\mathrm{I}}|\kappa_{2\omega}, n-2\rangle|^2, \\ \Delta &= \hbar\omega - E_b + E_g \\ &\quad - \langle b, n-1|H_{\mathrm{I}}\mathcal{G}_{2\omega}^{(\kappa_{2\omega})}H_{\mathrm{I}}|b, n-1\rangle, \end{aligned} \tag{20}$$

and the operators that appear are

$$\begin{aligned} P_{b,n} &= |b, n\rangle\langle b, n|, \\ \mathcal{G}_{2\omega}^{(\kappa_{2\omega})} &= \sum_{\kappa \neq \kappa_{2\omega}} \frac{|\kappa, n\rangle\langle\kappa, n|}{E_\kappa - E_g - 2\hbar\omega}. \end{aligned} \tag{21}$$

The $\Sigma$ symbol in $\mathcal{G}_{2\omega}^{(\kappa_{2\omega})}$ includes both the summation over the discrete states and the integration over the continuum states. As $\kappa_{2\omega}$ is in the continuum the exclusion $\kappa \neq \kappa_{2\omega}$ should be understood as taking the principal value of the integration part. Due to the normalization for the continuum states, the wavefunction of $|\kappa_{2\omega}\rangle$ in real space has the unit of $([\mathrm{energy}] \times [\mathrm{length}]^3)^{-1/2}$, and hence $\Gamma$ has the unit of energy as it should.

Following Armstrong et al.[15] (and correcting a sign error), the transition rate from the ground state $|g, n\rangle$ to the Fano eigenstate $|\chi\rangle$ can be obtained by Fermi's Golden Rule generalized for second-order processes[34]

$$R = \frac{\pi}{2\hbar}\left|\langle g, n|H_{\mathrm{I}} - H_{\mathrm{I}}^{(2)}|\chi\rangle\right|^2, \tag{22}$$

where $H_{\mathrm{I}}^{(2)} = H_{\mathrm{I}}\mathcal{G}_\omega^{(b)}H_{\mathrm{I}}$ is the Hamiltonian for the two-photon excitation through the non-resonant intermediate states. It is noteworthy that $H_{\mathrm{I}}$ is odd, so that the atomic wavefunctions $|g\rangle$ and $|\kappa_{2\omega}\rangle$ have the same parity, opposite to $|b\rangle$. By expanding the matrix element and utilizing this selection rule, one can show that

$$R = \frac{\pi}{2\hbar}\frac{|M(\Delta)|^2}{\Delta^2 + \Gamma^2}, \tag{23}$$

where $M(\Delta) = \langle g, n|H_{\mathrm{MPICS}}(\Delta)|\kappa_{2\omega}, n-2\rangle$ and

$$\begin{aligned} H_{\mathrm{MPICS}}(\Delta) =& H_{\mathrm{I}}P_{b,n-1}H_{\mathrm{I}} - H_{\mathrm{I}}^{(2)}\Delta \\ &- H_{\mathrm{I}}^{(2)}\mathcal{G}_{2\omega}^{(\kappa_{2\omega})}H_{\mathrm{I}}P_{b,n-1}H_{\mathrm{I}}. \end{aligned} \tag{24}$$

As $\mathcal{G}_\omega^{(j)}$ has units of $[\mathrm{energy}]^{-1}$, $H_{\mathrm{I}}^{(2)}$ and $H_{\mathrm{MPICS}}$ have units of energy and $[\mathrm{energy}]^2$, respectively. Whereas the first two terms in $H_{\mathrm{MPICS}}$ are second order in

$H_{\mathrm{I}}$, the third term is fourth order in $H_{\mathrm{I}}$ and involves non-resonant continuum states (through $\mathcal{G}_{2\omega}^{(\kappa_{2\omega})}$). Likewise, the energy shift of the resonance, the last term in the equation for $\Delta$ above, is second order in $H_{\mathrm{I}}$ and also involves $\mathcal{G}_{2\omega}^{(\kappa_{2\omega})}$, whereas the others are zeroth order. We neglect these higher order terms in our calculation below.

Now, one can use the identity $a_\omega|n\rangle = \sqrt{n}|n-1\rangle$ to eliminate the field component in the matrix elements. Noting that the electric field amplitude is $F = (n\hbar\omega/2\epsilon_0\epsilon_r L^3)^{1/2}$ and $n$ is very large, and defining $|k_{2\omega}\rangle = \sqrt{E_{\mathrm{H}}}|\kappa_{2\omega}\rangle$, we obtain Eq. (8) in the main text:

$$\Gamma = \pi E_{\mathrm{H}}|\langle b|d|k_{2\omega}\rangle|^2\left(\frac{F}{F_a}\right)^2, \tag{25}$$

(where the matrix element is dimensionless). $M(\Delta)$ may now be written

$$\begin{aligned} M(\Delta) =& E_{\mathrm{H}}^{1/2}\left(\frac{F}{F_a}\right)^2 \\ &\times \left(E_{\mathrm{H}}\langle g|d|b\rangle\langle b|d|k_{2\omega}\rangle - \Delta\langle g|dG_\omega^{(b)}d|k_{2\omega}\rangle\right) \\ &\propto \langle g|(d - dG_\omega^{(b)}d)|\Psi\rangle, \end{aligned} \tag{26}$$

where $G_\omega^{(b)}$ is given by Eq. (5) in the main text and $|\Psi\rangle$ is an un-normalized effective Fano state of the atom only,

$$|\Psi\rangle = |b\rangle + \theta|k_{2\omega}\rangle, \tag{27}$$

(shown in Fig. 5) where the dimensionless mixing parameter is given by $\theta = \Delta/(E_{\mathrm{H}}\langle b|d|k_{2\omega}\rangle)$, or

$$|\theta| = \Delta\sqrt{\frac{\pi}{E_{\mathrm{H}}\Gamma}}\frac{F}{F_a}, \tag{28}$$

The rate $R$ now has the form of Eq. (6) given in the main text:

$$R = R_0\frac{(\Delta + q\Gamma)^2}{\Delta^2 + \Gamma^2}, \tag{29}$$

where $q$ is given by Eq. (7) and $R_0$ is the two-photon background rate to the undressed continuum via the non-resonant intermediate states,

$$R_0 = \frac{\pi}{2\hbar}E_{\mathrm{H}}|\langle g|dG_\omega^{(b)}d|k_{2\omega}\rangle|^2\left(\frac{F}{F_a}\right)^4. \tag{30}$$

As mentioned in the main text, for $|b\rangle \equiv 2p$ in H, we find that $\langle g|d|b\rangle = 0.74$. Due to selection rules the continuum state may have angular momentum $l = 0$ or $l = 2$. We find that the background two-photon ionization rate, $R_0$, to the $l = 2$ continuum state is 30 times larger than the rate to the $l = 0$ continuum state, and hence we neglect the latter. Then $\langle b|d|k_{2\omega}\rangle = 0.38$, $\langle g|dG_\omega^{(b)}d|k_{2\omega}\rangle = 8.5$ and hence $\Gamma = 0.45 E_{\mathrm{H}}F^2/F_a^2$ and $q = -0.074 F_a^2/F^2$, so that the zero in the Fano lineshape is at $\Delta_0 = -q\Gamma = 0.033 E_{\mathrm{H}}$ (independent of field, and therefore robust against fluctuations) and the peak is at $\Delta_p = \Gamma/q = -6.1 E_{\mathrm{H}}F^4/F_a^4$.

In silicon, if we simply scale the hydrogen result as mentioned in the main text, then $E_{\mathrm{H}} = 40$ meV, so we expect the zero at $\Delta_0 = 1.4$ meV, and $F_a = 0.13$ MV/cm so with our experimental conditions $F$ is ~22 kV/cm and the peak is at $\Delta_p = -0.2$ meV. These conditions imply $\theta \approx 0.09$ at the zero and the peak rate when $\theta \approx -0.01$, and the corresponding dressed excited state wavefunctions $|\Psi\rangle$ for these values of $\theta$ are shown in Fig. 5. At zero detuning when $|\Psi\rangle = |b\rangle$, the photo-ionization is obviously purely step-wise via the discrete state using one-photon excitation. At very large detuning the excitation is purely to the continuum via two-photon excitation (where the intermediate state is off-resonant). At the zero (which is field independent) the amplitudes of the two contributions are comparable, so that the matrix elements with the ground state are equal and opposite. For the relatively small electric field we used, the wavefunction at peak ionization has a quite small component of the continuum and is dominated by the pathway through the bound state.

## Data availability

The data for this work are available without restriction at https://doi.org/10.5281/zenodo.3406204.

## Code availability

Mathematical algorithms for this work are available without restriction at https://doi.org/10.5281/zenodo.3406204.

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

## Acknowledgements

We acknowledge financial support from the UK Engineering and Physical Sciences Research Council (ADDRFSS, grant reference EP/H001905/1). The work at FELIX was performed as part of the research programme of the Stichting voor Fundamenteel Onderzoek der Materie (FOM), which is financially supported by the Nederlandse Organisatie voor Wetenschappelijk Onderzoek (NWO). Z.H. thanks the China National Science Fund for Distinguished Young Scholars (61625505) and the Sino-Russia International Joint Laboratory of Terahertz (18590750500).

## Author contributions

K.L.L. proposed the experiment. K.L.L. and B.R. performed the experiments. N.H.L., B.N. M., and K.L.L. provided the theory and analysed the data. N.V.A. provided samples. B.N. M., C.R.P., K.L.L., N.H.L., Y.A., and Z.H. wrote the paper.

## Competing interests

The authors declare no competing interests.
