## [Peer Review File · Nature Communications]

REVIEWER COMMENTS

Reviewer #1 (Remarks to the Author):

Litvinenko et al. demonstrate the multi-photon Fano effect using a THz laser applied to a phosphorus donor in silicon. Since the interference of single level and continuum that leads to the Fano line shape is induced here using the laser, then changes in the transmission spectrum can be controlled on a picosecond timescale. This could be very important, for example for readout of silicon donors in a quantum information processor.

I found the results to be very interesting and I feel that this work is significant, and will be of interest to those working on silicon devices. I therefore feel that the paper should move towards publication in Nature Communications. However, I also felt that at times the work was not as well presented as it could be. I am not sure how close the authors are to the length limit for the main part of the paper, but I felt there were places where an expansion of the text could help the reader. I will discuss these in more detail below.

Apart from these largely presentational issues, there is one main point that needs to be addressed before I can make a final recommendation. The authors claim a ps-switch on of the Fano resonance. For a quantum devices readout, one would want to do this switching repeatedly. Is it possible for the authors to show successive switching on and off of the transmission?

I now go through the paper with some further questions and comments:

1. Through the paper I felt that the theory was discussed too briefly, and this makes it hard to follow. (There is more detail in the supplement, but I feel the main paper presentation could be improved). On page 2, the interaction V is introduced - what is the origin of this interaction and what form does the interaction Hamiltonian take? Similarly, D is introduced but no detail of the kind of interaction this corresponds to is given.
2. Fig 2. Could the symbols E_p and $\hbar\omega$ be linked to the quantities in the caption? In particular, (c) shows a wavelength dependence according to the caption but are quoted as different frequencies in the figure itself.
3. Fig 2. Caption. (d) is mentioned but is not in the figure.
4. Page 2, col 2. It is quoted that MPICS has $q \propto 1/F^2$ *also*. Why use also here?
5. Page 2, col 2. "Enriched" with what? At this point the fact that phosphorus donors are being studied has not been mentioned in the main text.
6. Page 2, col 2. Here ket notation is used and set equal to a quantity that is not in ket notation. Both sides should be kets really.
7. Page 2-3. I was confused about the discussion of negative induced transmission where they say that this happens off resonance or for high intensity. I cannot see this in Fig 2(a) for high intensity. Can this be clarified?
8. Page 3. I think more detail is needed on what "reciprocal decay" is I do not think this term is widely known. Also it has the wrong sign in Fig 1.
9. Page 3. I found the discussion of the theory here confusing. It is better in the supplement but I think it could be improved here. For example, in the discussion of the standard Fano effect, they use

the state “ c ” in the text but it is “ k_{ω} ” in Fig. 1.

Reviewer #2 (Remarks to the Author):

This is a fine paper detailing how to create a Fano-like interference using multiphoton ionization. This is demonstrated experimentally, and theory is provided to explain the phenomenology in detail. I recommend publication. But there are a few small points of confusion (at least, to me) that I suggest that the authors take care of. Specifically:

1. I don't really understand the first sentence of the manuscript. What do widths have to do with anything. The only way an autoionizing state acquires a width is through its interaction with the continuum. Please modify and/or expand upon this to make it clearer.

2. At the end of the first paragraph, the term "shape index" is used and i didn't understand it until later in the paper when it was defined. I suggest that, in parentheses, after shape index, maybe (the Fano q -parameter) or something like that to make the connection with Ref. 1.

3. The cartoon of FIG. 1. is slightly confusing. In the Fano column of (a) the implication is that the autoionizing state $|a\rangle$ and the continuum state $|k\rangle$ have different energies which is, of course, not true. I wonder if it can be tweaked a bit to abnegate this incorrect impression, or at least explain in the caption that the vertical axis is not exactly energy.

Reviewer #3 (Remarks to the Author):

In this paper, the authors experimentally demonstrated Fano lineshapes by Multi-Photon-Induced Continuum Structure. The experiments were based on a platform they developed before with Giant multiphoton absorption (Nature Photon 12, 179–184 (2018)). Taking advantages of an increased Bohr radius of Si:P, the required intensity for MPICS is achievable by using a Free- Electron Laser. A Fano lineshape is experimentally observed as shown in Fig. 3b. The values of shape index q from the experiment and theory they developed have a good match.

However, I do not see a fundamental impactful advance here. The authors need to critically demonstrate the significance of their work, otherwise it is merely a “show-and-tell study”, unsuitable for Nature Communications.

Major concerns:

A.

“In the case of MPICS in particular, only one beam is needed and the shape index is controllable.”

In the manuscript, only a Fano line with $q=1.6$ was demonstrated.

Can the value of q be controlled in the proposed system?

B.

Two samples are prepared, one with natural isotopes and one with enriched isotopes.

In the manuscript, no further information is provided for the enriched ones.

In the supplementary, it is only mentioned that “non-resonant effects more pronounced”.

Some discussion in the manuscript may be needed.

C.

“open the possibility to tune donors in and out of invisibility to a single state-to-charge conversion readout beam.”

A direct comparison between the max and min transmission may clarify this claim.

The manuscript only shows the relative change in a.u.

Some Minor Concerns:

I.

Several typos of the indices of Figures & equations

For example,

The transients have been normalized (except in (d)) and offset vertically.

We fixed the time delay (at $t=15\text{ps}$) and swept the wavelength, as shown in Fig 3b.

These negative signals are also visible in Fig 2c, which was taken

The solid curves are the results of a global fit using Eqn 1 for the ionization

while A;B;C; t_R were different for each trace in Fig 2c.(supplementary)

II.

The comparison between theory and experiment is based on the value of F. A direct comparison of q and Γ may be more straightforward.

RESPONSE TO REFEREES

We thank the referees for their very careful reading and helpful comments. We have addressed all the issues they⁷⁸⁰ raise. In particular we have made major changes to the manuscript where requested, including new calculations and a detailed description of applications of the principles and primary discovery. We respond to each of the reviewers points in detail below (our responses in red).⁷⁸⁵

We hope the work is now suitable for publication in Nature Communications.

Yours,

Konstantin Litvinenko et al.

790

Reviewer 1

Litvinenko et al. demonstrate the multi-photon Fano effect using a THz laser applied to a phosphorus donor⁷⁹⁵ in silicon. Since the interference of single level and continuum that leads to the Fano line shape is induced here using the laser, then changes in the transmission spectrum can be controlled on a picosecond timescale. This could be very important, for example for readout of silicon donors in a quantum information processor.⁸⁰⁰

I found the results to be very interesting and I feel that this work is significant, and will be of interest to those working on silicon devices. I therefore feel that the paper should move towards publication in Nature Communications. However, I also felt that at times the work was not as well presented as it could be. I am not sure how close the authors are to the length limit for the main part of the paper, but I felt there were places where an expansion of the text could help the reader. I will discuss these in more detail below. **Agreed - see our response to this Referee's point 1 below.**⁸¹⁰

Apart from these largely presentational issues, there is one main point that needs to be addressed before I can make a final recommendation. The authors claim a ps-switch on of the Fano resonance. For a quantum devices readout, one would want to do this switching repeatedly. Is it possible for the authors to show successive switching on and off of the transmission? **Yes. We have already demonstrated ultrafast switch-on shown in Fig 2 , and we also already demonstrated successive repeats with laser repetition time of 40ns, which we now mention on line 548. This repetition speed is now emphasised on line 398. Furthermore, we have significantly expanded the general discussion of the applications in response to Referee 3.**⁸²⁰

I now go through the paper with some further questions and comments:⁸²⁵

1. Through the paper I felt that the theory was discussed too briefly, and this makes it hard to follow. (There is more detail in the supplement, but

730

735

740

745

750

755

760

765

770

775

I feel the main paper presentation could be improved). On page 2, the interaction V is introduced - what is the origin of this interaction and what form does the interaction Hamiltonian take? Similarly, D is introduced but no detail of the kind of interaction this corresponds to is given. **Following the general suggestion of the Referee we have moved a significant fraction of the theory from the Supplementary Materials to the main manuscript, and the remainder to the Methods section so that it will appear alongside the manuscript. We have also introduced section headings (according to the Nat Comm style), in order to signal the flow to the reader in a better way. In relation to the specific comments: we now explicitly define V as "the interaction between the discrete state and the continuum, V " and identify it as "the electron-electron configuration interaction in the helium example just given" - see line 86. We have greatly expanded the caption of Fig 1 to explain this more clearly.**

2. Fig 2. Could the symbols E_p and $\hbar\omega$ be linked to the quantities in the caption? In particular, (c) shows a wavelength dependence according to the caption but are quoted as different frequencies in the figure itself. **Agreed. We have removed the word "wavelength" and now explicitly mention $\hbar\omega$ and E_p in the caption.**
3. Fig 2. Caption. (d) is mentioned but is not in the figure. **Agreed. We have corrected this reference to a non-existent panel.**
4. Page 2, col 2. It is quoted that MPICS has $q \propto 1/F^2$ *also*. Why use also here? **We mean to stress that in LICS only Γ is controllable with field F , not q , but in MPICS both Γ and q are controllable. We have reworded the phrase - see the two sentences ending line 132. Although q and Γ are not independently controllable with F , this has experimental advantages because the window-position is robust against field fluctuations, as we mention on line 417.**
5. Page 2, col 2. Enriched with what? At this point the fact that phosphorus donors are being studied has not been mentioned in the main text. **Agreed. We now mention at this point the phosphorus doping - see line 154. Furthermore, in response to Reviewer 3 point 2 we have removed the isotopically enriched sample data from this manuscript.**
6. Page 2, col 2. Here ket notation is used and set equal to a quantity that is not in ket notation. Both sides should be kets really. **Agreed. We have changed this line 161 and the caption of Fig 5 accordingly.**

- 830 7. Page 2-3. I was confused about the discussion of
negative induced transmission where they say that
this happens off resonance or for high intensity. I
cannot see this in Fig 2(a) for high intensity. Can
this be clarified? **The referee is correct - we have**
835 **not shown data exhibiting induced absorption at**
high intensity on resonance - it happens at higher
intensity than shown on Fig 2a when $C \propto n_{c0}$ ex-
ceeds $B \propto n_{b0}$. We have modified the sentence to
avoid this confusion - see line 167.
- 840 8. Page 3. I think more detail is needed on what⁸⁹⁰
“reciprocal decay” is I do not think this term is
widely known. Also it has the wrong sign in Fig 1.
Agreed, the sign in Fig 1 has been corrected. A refer-
ence was given to reciprocal decay, but we have
845 **expanded the text to give more explanation and**
more rate equation theory detail where it is first
mentioned - see the sentences between line 195 and
Eqn 3.
- 850 9. Page 3. I found the discussion of the theory here
confusing. It is better in the supplement but I think
it could be improved here. For example, in the dis-
cussion of the standard Fano effect, they use the
state c in the text but it is k_ω in Fig. 1. **Regarding**
the difference between c and k_ω , we tried to distin-
855 **guish between the continuum and the states of dif-**
ferent energy within it (and we defined “ $|k_\omega\rangle \in |c\rangle$ ”
is the unmixed continuum state at energy $E_g + \hbar\omega$ ”
just above Eqn 5). We have now significantly ex-
860 **expanded the caption of Fig 1 to make this clear.**
In response to the general point that theory in the
supplement was more helpful, we have moved a sig-
nificant fraction to the main text (see our response
to this Referee’s point 1 above).

Reviewer 2

865 This is a fine paper detailing how to create a Fano-
like interference using multiphoton ionization. This is
demonstrated experimentally, and theory is provided to
explain the phenomenology in detail. I recommend pub-
lication. But there are a few small points of confusion (at⁹²⁰
870 least, to me) that I suggest that the authors take care of.
Specifically:

- 875 1. I don’t really understand the first sentence of the
manuscript. What do widths have to do with any-⁹²⁵
thing. The only way an autoionizing state acquires
a width is through its interaction with the contin-
uum. Please modify and/or expand upon this to
make it clearer. **To answer the referee’s question**
about widths: the “discrete” (autoionising) state
880 **involved in a Fano resonance does not have to be**

infinitely sharp before interaction with the contin-
uum, all that is necessary is that its natural spectral
width is much smaller than the width of the contin-
uum. But the referee is correct that the generality
of our statement was unnecessary and could lead
to confusion, and therefore we have now simplified
it - see line 26.

2. At the end of the first paragraph, the term “shape
index” is used and i didn’t understand it until later
in the paper when it was defined. I suggest that,
in parentheses, after shape index, maybe (the Fano
 q -parameter) or something like that to make the
connection with Ref. 1. **Agreed. We have followed**
the referee’s suggestion - see line 44 - and also men-
tioned q in the abstract where the shape index is
mentioned.
3. The cartoon of FIG. 1. is slightly confusing. In
the Fano column of (a) the implication is that the
autoionizing state $|a\rangle$ and the continuum state $|k\rangle$
have different energies which is, of course, not true.
I wonder if it can be tweaked a bit to abnegate
this incorrect impression, or at least explain in the
caption that the vertical axis is not exactly energy.
We agree that Fig 1a was confusing, and to an-
swer referee’s comment (and others by Referee 1)
we have now significantly expanded its caption in
order to explain it clearly. Please note: while the
detuning must be small for a Fano resonance to be
observed, in general the final continuum state in
the Fano theory $|k_\omega\rangle$ and the auto-ionising state
 $|a\rangle$ have different energy, the difference being $\hbar\Delta$,
and the figure indicates the situation when $\Delta > 0$
in Eqn 1 ($|k_\omega\rangle$ above $|a\rangle$). We now state this ex-
PLICITLY in the caption.

Reviewer 3

915 In this paper, the authors experimentally demon-
strated Fano lineshapes by Multi-Photon-Induced Con-
tinuum Structure. The experiments were based on a
platform they developed before with Giant multiphoton
absorption (Nature Photon 12, 179184 (2018)). Taking
advantages of an increased Bohr radius of Si:P, the re-
quired intensity for MPICS is achievable by using a Free-
Electron Laser. A Fano lineshape is experimentally ob-
served as shown in Fig. 3b. The values of shape index q
from the experiment and theory they developed have a
good match.

However, I do not see a fundamental impactful ad-
vance here. The authors need to critically demonstrate
the significance of their work, otherwise it is merely a
show-and-tell study, unsuitable for Nature Communica-
tions. **The fundamental advances here are several and**
major. Here we produce the experimental demonstration

of laser-induced Fano interference with an effect that does not require extreme intensity, for the first time to our knowledge. The idea of the Fano line-shape, which carries both constructive and destructive interference (both a peak and a window), is extremely important because it gives structure to an otherwise flat continuum (line 33), and this allows a great variety of effects in many different disciplines, such as those we illustrated with Refs [2–14] (see first paragraph). To our knowledge, these and all other previous demonstrations of control over q are either: “permanent” influence over the design/choice of atom; slow control with temperature/magnetic field; or with extremely intense, non-perturbative drive lasers (or the control offered is only over the width or centre-frequency, not the shape). As we already stated - see the abstract, the introduction, and in particular line 132 - MPICS offers ultrafast control over q , in essentially any atom, and with relatively weak beams. This idea was first proposed by Armstrong in 1975 [15], which spawned a citation tree that includes hundreds of papers with theoretical developments, **but has never actually been realized until this work**. Our crucial discovery is that MPICS with q near unity is achievable with weak beams so long as the coupling beam frequency is low (i.e THz). Although we concentrated our discussion in the first version on the silicon donors of particular interest to us, we now emphasise the more general nature of this discovery at the end of the first paragraph, and in a new final paragraph. The advance here is not only experimental but also theoretical - we show how to calculate the MPICS q using a new variant of implicit summation, that has allowed us to model our results very well, with no adjustable parameters.

In response to the Referee’s request for a specific application, we have significantly expanded the discussion of the applications, from line 404 to the end of the main text, including the new Fig 4. In order to emphasise the wide applicability and the potential for broad impact in atomic physics (etc), we give an example (see last paragraph) with one of the most well studied atomic vapours, rubidium, and we modified the last sentence of the abstract to reflect this. We show that contrary to widely held beliefs (as just mentioned, Ref [15] is at the top of a tree of hundreds of papers) we discovered that ICS is straightforwardly accessible so long as THz coupling beams (from available laser diodes) are used.

Major concerns:

1. “In the case of MPICS in particular, only one beam is needed and the shape index is controllable.” In the manuscript, only a Fano line with $q=1.6$ was demonstrated. Can the value of q be controlled in the proposed system? Yes, certainly, this is our central claim. It can be controlled by choice of atom or system, which varies the matrix elements, and it can be controlled with laser field, as shown

by Eqn 7. We have achieved very good agreement with the experiment (results from fitting of Figs 3b & c) and the MPICS theory using a model *with no adjustable parameters* (now emphasised explicitly on line 351) - it is a simple scaling of mass and permittivity from hydrogen as we stated on line 347. This model shows that all that is needed to change q is to change the electric field strength of the light pulse.

2. Two samples are prepared, one with natural isotopes and one with enriched isotopes. In the manuscript, no further information is proved for the enriched ones. In the supplementary, it is only mentioned that “non-resonant effects more pronounced”. Some discussion in the manuscript may be needed. We agree with the referee that the data from the isotopically enriched sample is not central to the point of this manuscript, and as a result we have removed it to be published elsewhere.
3. “open the possibility to tune donors in and out of invisibility to a single state- to-charge conversion readout beam.” A direct comparison between the max and min transmission may clarify this claim. The manuscript only shows the relative change in a.u. Agreed. We have now put the full scale on the y-axes of Figs 3b&c as the Referee suggests.

Some Minor Concerns:

1. Several typos of the indices of Figures & equations, for example, “The transients have been normalized (except in (d)) and o set vertically.” Agreed - see response to Referee 1 point 3. “we fixed the time delay (at $t=15$ ps) and swept the wavelength, as shown in Fig 3b.” Agreed, we have changed “wavelength” to “photon energy” - line 176. “These negative signals are also visible in Fig 2c, which was taken”. Agreed, we have changed “Fig 2c” to “Fig 2b” - line 180. “The solid curves are the results of a global fit using Eqn 1 for the ionization” Agreed, we have inserted the phrases “(blue line)” and “(orange line)” to identify which curve corresponds with Eq 1 and which corresponds to the bleaching/Lorentzian (see caption Fig 3). “while A;B;C; tR were different for each trace in Fig 2c.(supplementart)” Agreed - we have removed the repetition of “in Fig 2c” (see line 604).
2. The comparison between theory and experiment is based on the value of F . A direct comparison of q and Γ may be more straightforward. The experimental field amplitude has a potentially large systematic error due to the difficulty of measuring the temporal pulse profile, and it is therefore somewhat misleading to produce a “theoretical” q

1040

and Γ for this field. This was our reason for working backwards from the much more experimentally reliable q value. Nevertheless we now include a straightforward comparison of q and Δ_0 (the window position). We have reworded the discussion

of uncertainties in order to make the field comparisons straightforward also, while emphasizing the sources of uncertainty. See line 357 to line 368.

REVIEWERS' COMMENTS

Reviewer #1 (Remarks to the Author):

The authors have done a good job in responding to my previous report. I think they have dealt with everything thoroughly, and so I am now happy to recommend publication.

Brendon Lovett

Reviewer #3 (Remarks to the Author):

The authors properly addressed the questions raised previously.

The authors provided detailed information to show that their observations are indeed based on “laser-induced Fano interference that does not require extreme intensity”, which was previously proposed (ref 15) but not realised.

This clarifies my previous concerns about the impact of the manuscript.

Therefore, I recommend the acceptance of the paper.